

PeerJ Hubs
Published on behalf of



International Association for Biological Oceanography
IABO

# Baseline dynamics of Symbiodiniaceae genera and photochemical efficiency in corals from reefs with different thermal histories

Crystal J. McRae[1,2], Shashank Keshavmurthy[3], Hung-Kai Chen[1], Zong-Min Ye[1], Pei-Jie Meng[1,4], Sabrina L. Rosset[5], Wen-Bin Huang[6], Chaolun Allen Chen[3], Tung-Yung Fan[1,7] and Isabelle M. Côté[2]

[1] Department of Planning and Research, National Museum of Marine Biology & Aquarium, Checheng, Pingtung, Taiwan
[2] Department of Biological Sciences, Simon Fraser University, Burnaby, British Columbia, Canada
[3] Biodiversity Research Center, Academia Sinica, Nangang, Taipei, Taiwan
[4] Graduate Institute of Marine Biology, National Dong Hwa University, Checheng, Pingtung, Taiwan
[5] School of Biological Sciences, Victoria University of Wellington, Wellington, New Zealand
[6] Department of Natural Resources and Environmental Studies, National Dong Hwa University, Shoufeng, Hualien, Taiwan
[7] Department of Marine Biotechnology and Resources, National Sun Yat-sen University, Kaohsiung, Taiwan

## ABSTRACT

Ocean warming and marine heatwaves induced by climate change are impacting coral reefs globally, leading to coral bleaching and mortality. Yet, coral resistance and resilience to warming are not uniform across reef sites and corals can show inter- and intraspecific variability. To understand changes in coral health and to elucidate mechanisms of coral thermal tolerance, baseline data on the dynamics of coral holobiont performance under non-stressed conditions are needed. We monitored the seasonal dynamics of algal symbionts (family Symbiodiniaceae) hosted by corals from a chronically warmed and thermally variable reef compared to a thermally stable reef in southern Taiwan over 15 months. We assessed the genera and photochemical efficiency of Symbiodiniaceae in three coral species: *Acropora nana*, *Pocillopora acuta*, and *Porites lutea*. Both *Durusdinium* and *Cladocopium* were present in all coral species at both reef sites across all seasons, but general trends in their detection (based on qPCR cycle) varied between sites and among species. Photochemical efficiency (*i.e.*, maximum quantum yield; $F_v/F_m$) was relatively similar between reef sites but differed consistently among species; no clear evidence of seasonal trends in $F_v/F_m$ was found. Quantifying natural Symbiodiniaceae dynamics can help facilitate a more comprehensive interpretation of thermal tolerance response as well as plasticity potential of the coral holobiont.

Corresponding authors
Crystal J. McRae,
crystal.j.mcrae@gmail.com
Tung-Yung Fan,
tyfan@nmmba.gov.tw

## INTRODUCTION

The health and persistence of corals in the Anthropocene are most highly threatened by climate change (*Hoegh-Guldberg et al., 2017*; *Hughes et al., 2017*), in particular ocean warming and marine heatwaves (*Spalding & Brown, 2015*; *Leggat et al., 2019*). Elevated seawater temperatures can lead to oxidative stress in corals (*Oakley & Davy, 2018*), which prompts the expulsion of the coral's symbiotic dinoflagellate algae (family Symbiodiniaceae), from which scleractinian (*i.e.*, reef-building) corals typically obtain most of their energy (*Yellowlees, Rees & Leggat, 2008*; *Van Oppen & Lough, 2018*). This physiological response leads to coral bleaching and may result in mortality, which can have detrimental effects on reef ecosystems and coral-dependent communities (*e.g.*, phase shifts to algal-dominated states (*Ostrander et al., 2000*; *Vaughan et al., 2021*); homogenization of fish populations *Richardson, Graham & Hoey, 2020*).

The effect of elevated temperature on corals, however, is not uniform among species and sites. Corals with 'weedy' life-history traits (*e.g.*, *Pocillopora* spp.), which typically have a brooding reproductive strategy and are recruitment pioneers, appear to fare better under elevated temperatures than 'competitive' corals (*e.g.*, *Acropora* spp.), which are fast-growing broadcast spawners, but neither performs as well as 'stress-tolerant' corals (*e.g.*, massive *Porites* spp.), which are slow-growing broadcast spawners (*Darling, McClanahan & Côté, 2013*; *Kubicek et al., 2019*). In addition, reef site characteristics can also affect coral thermal tolerance (*Camp et al., 2018*; *Burt et al., 2020*). Reef sites that have shown higher resistance to bleaching include some that experience chronic disturbance (*e.g.*, *Guest et al., 2016*), a wide seasonal range of temperatures –including chronic high maximum temperatures (*e.g.*, *Riegl & Purkis, 2012*) and/or highly variable thermal regimes (*e.g.*, *Wyatt et al., 2020*). The primary mechanisms that allow some corals to persist and prosper are likely underpinned on a fundamental level by the capacity for effective and consistent energy acquisition. Here we focus on coral performance in relation to the genera and photochemical efficiency of the coral's symbiotic algae.

The symbiotic relationship between scleractinian corals and Symbiodiniaceae has allowed coral reefs to flourish in nutrient-poor marine environments due to the efficient transfer of photosynthetic products from the symbiont to the coral host (*Muscatine, 1990*; *Roth, 2014*). Corals can associate with a range of Symbiodiniaceae genera, often more than one at a time (*Baker, 2003*; *Silverstein, Correa & Baker, 2012*), and the genus of Symbiodiniaceae can influence the host's bleaching resistance (*Berkelmans & van Oppen, 2006*). For example, the genus *Durusdinium* (formerly known as clade D) has been associated with high bleaching resistance under both heat and cold stress (*Stat & Gates, 2011*; *Silverstein, Cunning & Baker, 2017*), but typically at the cost of slower coral growth compared to the more thermally sensitive genus *Cladocopium* (formerly known as clade C) (*Jones & Berkelmans, 2010*; *LaJeunesse et al., 2018*). Symbiodiniaceae genus composition can vary spatially, with corals in more thermally extreme sites generally being more likely to host relatively more *Durusdinium* than other genera (*Oliver & Palumbi, 2009*; *Keshavmurthy et al., 2012*; but see *Howells et al., 2020* for an example of thermally tolerant *Cladocopium*). Further, species within a genus of Symbiodiniaceae can also show

different responses to high temperature, which can vary based on coral host-algal symbiont pairing (*Hoadley et al., 2019*).

In general, a challenge to pinpointing potential mechanisms of thermal tolerance in corals is the lack of long-term, *in situ* physiological/molecular data under unstressed conditions (*e.g.*, relatively poor understanding of seasonal trends and natural fluctuations). Such data are needed to put monitoring studies (*e.g.*, pre- and post-bleaching surveys) and short-term laboratory experiments in context. This is especially pertinent when assessing the potential role of coral traits and reef site characteristics (*e.g.*, temperature regime) on the acquisition of thermal resistance and resilience.

We assessed the seasonal dynamics of algal symbionts hosted by corals from a chronically warmed and thermally variable reef compared to a thermally stable reef in southern Taiwan over 15 months. We examined the dynamics of Symbiodiniaceae genus associations and photochemical efficiency across three common coral species with different life-history traits: *Acropora nana*, *Pocillopora acuta*, and *Porites lutea*, at both sites. We predicted that in the absence of large thermal anomalies, Symbiodiniaceae genus/genera associations would remain stable across seasons but would show site- and species-specific differences due to the distinct thermal regimes of the reef sites and species life-histories, respectively. We predicted that photochemical efficiency would vary across seasons, but that seasonal trends among species would be similar.

## MATERIALS & METHODS

This study was initially reported in *McRae (2021)* in partial fulfillment of PhD requirements.

### Study sites

The two study sites, Outlet reef (21.931°E, 120.745°N) and Wanlitung reef (21.955°E, 120.766°N), are located in southern Taiwan (Fig. S1). Outlet reef is situated within Nanwan Bay and has a highly variable thermal regime (daily temperature variation ∼1.5−3.2 °C (*Carballo-Bolaños et al., 2019*)) due to internal tide-induced upwelling (*Jan et al., 2004*; *Keshavmurthy et al., 2019*). This reef has been chronically warmed for over three decades due to the warm water discharge from an adjacent nuclear power plant, and therefore experiences higher summer temperatures than other reefs in southern Taiwan (*Hung, Huang & Shao, 1998*; *Keshavmurthy et al., 2014*), which has led to acclimatization/adaptation in some corals and their algal symbionts (*e.g.*, *Mayfield, Fan & Chen, 2013*; *Carballo-Bolaños et al., 2019*). Wanlitung reef is situated on the western coast of the Hengchun peninsula (∼8 km northwest of Outlet reef) and has a relatively stable thermal regime throughout the year (daily temperature variation ∼0.7−1.6 °C (*Carballo-Bolaños et al., 2019*)). Both Outlet and Wanlitung reefs have been, and continue to be, impacted by anthropogenic stressors (*e.g.*, tourism and nutrient pollution (*Meng et al., 2008*; *Liu et al., 2012*)). Live coral cover in 2010 was ∼55% at Outlet reef (*Keshavmurthy et al., 2014*), and ∼17% at Wanlitung reef (*Kuo et al., 2012*).

### Monitoring abiotic conditions: temperature and nutrients

A temperature logger (HOBO pendant UA-002-08; Onset Computer Corporation, Bourne, MA, USA) was deployed at ∼3 m depth at each site and recorded seawater temperature

every 10 min from June 2018 to August 2019. Some temperature data collected in this study have been published as supplementary information for a complementary laboratory experiment using the same sites and coral species (see *McRae et al., 2022*). Water chemistry was monitored monthly at each site over approximately the same period following the protocols outlined in *Meng et al. (2008)*. Parameters measured included 5-day biological oxygen demand ($BOD_5$), nitrite ($NO_2^-$), nitrate ($NO_3^-$), ammonia ($NH_3$), and phosphate ($PO_4^{3-}$). Water samples were taken monthly from the same location and depth ($\sim$3 m) at each site; the two study sites were sampled on the same day and within 1 h of each other.

### Field collection

Ten coral colonies of each of the three coral species *A. nana*, *P. acuta*, and *P. lutea*, situated at $\sim$2–4 m depth, were tagged at each reef in May 2018. The same colonies were sampled five times (dates refer to Outlet and Wanlitung reefs, respectively): summer 2018 (July 30 and August 1), fall 2018 (October 25; both sites), winter 2019 (February 22 and 21), spring 2019 (April 27 and 23), and summer 2019 (August 29 and 27) (Kenting National Park collection permit numbers: 1080000091, 1090006457). In the rare instances where a colony had died or could not be relocated, a new colony was tagged and sampled (see Tables S1 and S2). Sampling at each site was conducted between $\sim$10:00 to 12:00. Three small pieces ($\sim$2 cm in length) from each colony were removed at each time point using shears (*A. nana, P. acuta*) or a hammer and chisel (*P. lutea*). Colonies of each species were situated at least 5 m away from each other (minor exceptions to this rule are noted in Tables S1 and S2) and intraspecific colony size was similar across sites. Despite repeated sampling on the same colonies, there was no evidence of long-term sampling damage to colonies; indeed, tissue had regrown over the wound sites between time points and there was no visual evidence of disease (C McRae, pers. obs. 2018 & 2019).

### DNA extractions and identification of Symbiodiniaceae genera

A subset of six colonies per species from each site was selected for Symbiodiniaceae analysis (see Tables S3 and S4) across each of the five time points assessed in this study. Colonies in the subset were selected if they were: (1) situated at least 5 m away from an adjacent colony of the same species, and (2) were collected at each seasonal time point across the study (minor exceptions noted in Table S4). While we acknowledge that exclusion of colonies that were not collected at each time point may present a subtle survival bias, we chose this approach because our assessment of seasonal trends necessitated data from each time point. Symbiodiniaceae analyses were successfully conducted on this subset for each season, with the exception of *P. lutea* at Outlet (only $n = 5$ colonies/season) and *P. acuta* (Outlet reef: only $n = 5$ colonies/season; Wanlitung reef: only $n = 2$ colonies/season). DNA extraction was accomplished using a salting-out method modified from *Ferrara et al. (2006)* based on the protocol described in *Keshavmurthy et al. (2020)*. In brief, coral tissue was lysed overnight in a two mL Eppendorf tube with 200 $\mu$L of lysis buffer (0.25 M Tris, 0.05 M EDTA at pH 8.0, 2% sodium dodecyl sulfate and 0.1 M NaCl) and 10 $\mu$L of 10 mg/mL proteinase E at 55 °C in a water bath. Then, NaCl (210 $\mu$L at 7 M) was added to the lysed tissue, and the sample was mixed by inverting the tube. The solution

was subsequently transferred to a 2 mL collection tube with a DNA spin column (Viogene, USA) and centrifuged at 8000 rpm for 1 min. The lysate was washed twice with 500 μL of ethanol (70%) by centrifuging at 8,000 rpm for 1 min during each step, with an additional centrifugation step at 8,000 rpm for 3 min to dry the spin column. The column was dried at 37 °C for 15 min, then the DNA was eluted with 50 μL of preheated (65 °C) 1X TE buffer, and was centrifuged at 15,000 g for 3 min. The quality of genomic DNA was assessed using a 1% agarose gel. The concentrations of genomic DNA were determined using NanoDrop 2000 (Thermal Scientific, Waltham, MA, USA).

The presence or absence of *Cladocopium* sp. and *Durusdinium* sp. in coral samples was determined using the primer pairs for *Cladocopium* and *Durusdinium*-specific ITS1 in qPCR assays (*Ulstrup & Van Oppen, 2003*). The detection cut-off cycle was set to 35 to avoid false positives caused by the formation of non-specific fluorescence (see *Mieog et al., 2007*). Each qPCR reaction contained 2X Fast Start Universal SYBR Green Master (ROX), 100 nM Symbiodiniaceae universal forward primer and *Cladocopium* or *Durusdinium*-specific reverse primer, 2.0 μL DNA template (similar concentration within species and within seasons), and deionized sterile water to a total volume of 10 μL. Based on the protocol described in *Keshavmurthy et al. (2022)*, amplifications were carried out on an StepOne Plus real-time PCR instrument (Applied Biosystems, Waltham, MA, USA) with thermal cycling conditions consisting of a denaturation step at 95 °C of 10 min followed by 35 two-step cycles of 15 s at 95 °C and 1 min at 60 °C. At the end of each run, a melt curve generated by temperature elevation from 60 °C to 95 °C in 0.5 °C increments each 5 s for 70 cycles was included to ensure that only target sequences were amplified. All qPCR reactions were run in triplicate (technical replicates), as was a no-template control (NTC) with ddH2O. It is important to note that specific quantification of the composition of each Symbiodiniaceae genus (*i.e.,* dominant *vs.* background genus) was not conducted in this study. Instead, we provide a coarse assessment of the presence/absence of Symbiodiniaceae upon which we hope future research can investigate in more detail.

## Maximum quantum yield ($F_v/F_m$) and sample fixation

Coral fragments were immediately transported to the research facilities of the National Museum of Marine Biology and Aquarium (Pingtung County, Taiwan) for sampling and photochemistry measurements. Corals were held in coolers filled with seawater from the source reef during transportation; transit duration was approximately 10 min (Wanlitung reef) or 20 min (Outlet reef). One fragment from each colony was dark adapted for 30 min and three replicate measurements of maximum quantum yield ($F_v/F_m$), a measure of photochemical efficiency (*Jones et al., 2000*), were taken using a diving PAM (Heinz Walz GmbH, Germany; settings: saturation pulse intensity = 8, measurement light intensity = 8, gain = 2, damp = 2). Measurements of $F_v/F_m$ were undertaken at our lab facilities (rather than in the field) to allow for effective dark adaptation, and because accurate *in situ* measurements would have been challenging during summer sampling due to the high wave action from typhoons and tropical storms commonly affecting the waters surrounding Taiwan. Another fragment from each colony was placed in ethanol for Symbiodiniaceae genera assessment. The last fragment from each colony was placed in liquid nitrogen for
lipid assessment, but due to an error in sample storage (*i.e.,* stored at $-20\,°C$, instead of $-80\,°C$) we have omitted these data due to concerns of possible sample degradation. We do, however, think it is important to mention our error here as a cautionary tale; see *Vega Thurber et al. (2022)* for a recent and comprehensive review of optimal sample fixation and storage protocols.

## Statistical analysis

We used linear mixed-effects models (and post-hoc pairwise comparisons) to assess (1) the effect of reef site and season, and their interaction, on water temperature (with month as a random effect to mitigate temporal autocorrelation), and (2) the effect of reef site, season, and coral species, and their interactions, on $F_v/F_m$ (with colony as a random effect to account for repeated sampling). Linear mixed-effects model assumptions were visually checked by plotting model residuals and through calculation of generalized variance inflation factors; $GVIF^{(1/(2*df)}$ was used to calculate generalized variation inflation factors as this approach is more suitable for categorical variables. We report the presence/absence of *Cladocopium* and *Durusdinium* detected in coral colonies at each site across seasons for each species. We used either a Student's *t*-test or a Mann–Whitney U test (depending on data distribution) to assess difference between reef sites for each water chemistry parameter; model assumptions of normality and equal variance were tested using Shapiro–Wilk and Levene's tests, respectively. All analyses were conducted in R (*R Core Team, 2022*) using the packages: lme4 (*Bates et al., 2015*), lmerTest (*Kuznetsova, Brockhoff & Christensen, 2017*), emmeans (*Length, 2022*), car (*Fox & Weisberg, 2019*), and lubridate (*Grolemund & Wickham, 2011*). All data and R scripts used in our analyses are publicly available on GitHub at: https://github.com/CJ-McRae/McRae-et-al_Peer-J-submission, and on Zenodo at https://doi.org/10.5281/zenodo.7762107.

# RESULTS

## Reef site temperature and nutrients

The mean temperature ($\pm$ SD) at Outlet and Wanlitung reefs over the study period of June 2018-August 2019 was $27.9 \pm 1.8\,°C$ and $27.8 \pm 1.7\,°C$, respectively (Figs. 1A & 1B). Mean daily temperature was higher at Outlet than Wanlitung reef in winter (linear mixed-effects model, post-hoc pairwise comparison; $t = 3.59, p = 0.01$) and spring ($t = 7.61, p < 0.001$) (Fig. 1C). Outlet reef had consistently larger daily temperature ranges ($2.85 \pm 1.25\,°C$) than Wanlitung ($1.27 \pm 0.59\,°C$) across each season (all $t \geq 5.12$, all $p < 0.001$) (Fig. 1D). Outlet reef had higher maximum temperatures across all seasons (all $t \geq 5.69$, all $p < 0.001$), with the exception of fall 2018 when no difference was found between sites (Fig. 1E). Lower temperature minima were also found at Outlet reef in summer 2018 ($t = -11.86, p < 0.001$) and summer 2019 ($t = -8.22$, all $p < 0.001$) (Fig. 1F). Water chemistry parameters did not differ between reef sites (Mann Whitney U tests, all $W \leq 79$, all $p > 0.05$ for $BOD_5$, $NO_2^-$, $NH_3$, and $PO_4^{3-}$; t-test, $t = 0.46, p > 0.05$ for $NO_3^-$) (Table S5). In brief, the means ($\pm$ SD) for Outlet and Wanlitung reefs, respectively, were $1.0 \pm 0.4$ and $1.1 \pm 0.6$ for $BOD_5$, $0.006 \pm 0.015$ and $0.003 \pm 0.005$ for $NO_2^-$, $0.017 \pm 0.007$ and

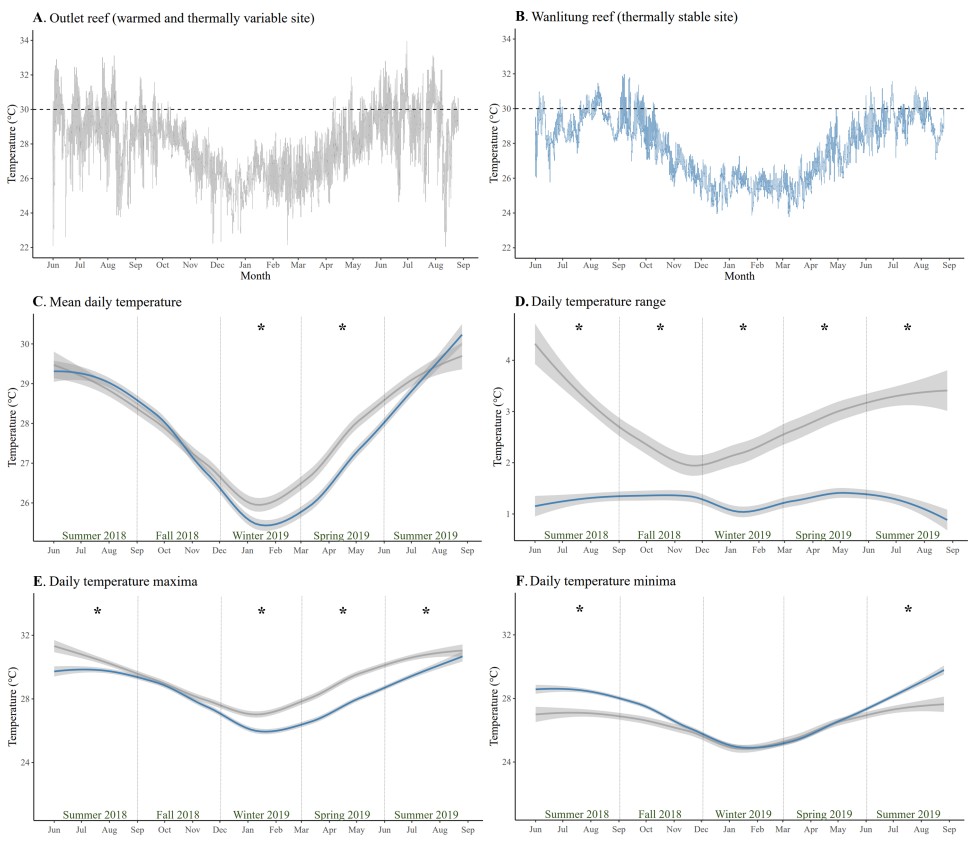

**Figure 1** **Temperature patterns from June 2018 to July 2019 at two study reefs in southern Taiwan.**
Daily temperature at (A) the warmed and thermally variable reef (Outlet reef; grey) and (B) the thermally
stable reef (Wanlitung reef; blue). The dashed horizontal lines show a reference temperature of 30 °C to
facilitate a fast visual comparison of temperature trends between sites. (C) Daily mean temperature, (D)
daily temperature range, (E) daily temperature maxima, and (F) daily temperature minima at each reef;
Outlet reef (grey) and Wanlitung reef (blue). The lines are smoothed averages and shaded areas represent
95% confidence intervals. Asterisks indicate season-specific temperature difference between sites.

$0.018 \pm 0.007$ for $NH_3$, $0.006 \pm 0.005$ and $0.011 \pm 0.028$ for $PO_4^{3-}$, and $0.017 \pm 0.014$
and $0.014 \pm 0.010$ for $NO_3$.

## Symbiodiniaceae associations

All corals hosted a combination of both Symbiodiniaceae genera across all seasons at both
sites (Figs. 2–4). In general, the detection of the genus of Symbiodiniaceae (*i.e.,* based
on qPCR amplification cycle) showed site- and species-specific patterns that remained
relatively consistent across seasons. Colonies of *A. nana* at Outlet reef (*i.e.,* the warmed
and thermally variable reef) showed the presence of both *Cladocopium* and *Durusdinium*
starting at ∼15 qPCR cycles, whereas *A. nana* from Wanlitung reef (*i.e.,* the thermally stable
reef) showed the presence of *Cladocopium* at ∼15 qPCR cycles and *Durusdinium* at ≥ 30
qPCR cycles (Fig. 2). In *P. acuta*, *Durusdinium* was detected earlier (∼15 qPCR cycles) than
*Cladocopium* (≥ 25 qPCR cycles) in colonies from Outlet reef, but the opposite pattern
was observed for colonies from Wanlitung reef (*i.e., Cladocopium* detected at ∼15 qPCR

cycles and *Durusdinium* detected at $\geq 25$ qPCR cycles) (Fig. 3). In *P. lutea, Cladocopium* was detected at ~15 qPCR cycles in colonies from both reef sites, but *Durusdinium* was detected earlier ($\geq 20$ qPCR cycles) in colonies from Outlet reef than Wanlitung reef ($\geq 30$ qPCR cycles) (Fig. 4). Coral samples for the summer 2019 time point were not analyzed due to subsequent COVID-19 restrictions to laboratory access.

## Maximum quantum yield ($F_v/F_m$)

Intraspecific differences in $F_v/F_m$ between reef sites occurred periodically in some seasons, with colonies from Outlet reef (*i.e.*, the warmed and thermally variable reef) having higher $F_v/F_m$ than colonies from Wanlitung reef (*i.e.*, the thermally stable reef) (Figs. 5A–5C). This was the case for *A. nana*, in fall 2018 and summer 2019 (linear mixed-effects model, post hoc pairwise comparisons; both $t \geq 2.47$, $p \leq 0.01$), for *P. acuta* in fall 2018 and spring 2019 (both $t \geq 3.00$, $p \leq 0.003$), and for *P. lutea* in summer 2018 only ($t = 2.06$, $p = 0.04$).

Interspecific differences in $F_v/F_m$ showed similar trends at both reef sites. Colonies of *P. acuta* and *A. nana* did not differ in $F_v/F_m$ in any season, and both species periodically had higher $F_v/F_m$ than *P. lutea* (mean $\pm$ SD across all seasons; Outlet reef, *A. nana*: $0.73 \pm 0.03$, *P. acuta*: $0.75 \pm 0.03$, *P. lutea*: $0.63 \pm 0.04$; Wanlitung reef, *A. nana*: $0.71 \pm 0.04$, *P. acuta*: $0.71 \pm 0.03$, *P. lutea*: $0.63 \pm 0.03$) (Figs. 5A–5C). At Outlet reef, both *P. acuta* and *A. nana* had higher $F_v/F_m$ than *P. lutea* in fall 2018, winter 2019, spring 2019, and summer 2019 (linear mixed-effects model, post-hoc pairwise comparisons; all $t \geq 2.58$, all $p < 0.03$) but only *P. acuta* was higher than *P. lutea* in summer 2018 ($t = 4.29$, $p = 0.001$). At Wanlitung reef, both *P. acuta* and *A. nana* had higher $F_v/F_m$ than *P. lutea* in summer 2018, winter 2019, spring 2019, and summer 2019 (all $t \geq 3.23$, all $p \leq 0.004$), but only *P. acuta* was higher than *P. lutea* in summer 2018 ($t = 4.00$, $p < 0.001$). There was no difference in $F_v/F_m$ among species at Wanlitung reef in fall 2018.

Overall, there were no clear seasonal trends in $F_v/F_m$ but some variation over time was observed in each species at each site (Figs. 5A–5C). In *A. nana*, $F_v/F_m$ was higher in spring 2019 than summer 2018 at Outlet reef (linear mixed-effects model, post-hoc pairwise comparisons; $t = 3.61$, $p = 0.003$), whereas at Wanlitung reef $F_v/F_m$ was higher in spring 2019 than all other seasons (all $t \geq 3.67$, all $p \leq 0.003$). In *P. acuta*, $F_v/F_m$ was higher in spring 2019 than summer 2019 at Outlet reef ($t = 3.56$, $p = 0.004$), in contrast to Wanlitung reef where $F_v/F_m$ was only higher in winter 2019 compared to fall 2018 ($t = 3.57$, $p = 0.004$). In *P. lutea*, $F_v/F_m$ at Outlet reef was lower in summer 2019 than all other seasons except for spring 2019, and was higher in fall 2018 compared to spring 2019 (all $t \geq 2.99$, all $p \leq 0.024$), whereas at Wanlitung reef $F_v/F_m$ was higher in fall 2018 and spring 2019 than in summer 2019 (both $t \geq 3.34$, both $p \leq 0.035$).

## DISCUSSION

Our comparison of algal symbiont dynamics in *A. nana*, *P. acuta*, and *P. lutea* colonies at a warmed and thermally variable reef (Outlet reef) and a thermally stable reef (Wanlitung reef) showed site- and species-specific patterns. In general, *Durusdinium* was detected earlier in the qPCR cycle than *Cladocopium* in colonies from the warmed and thermally variable reef than at the thermally stable reef. Symbiodiniaceae associations remained

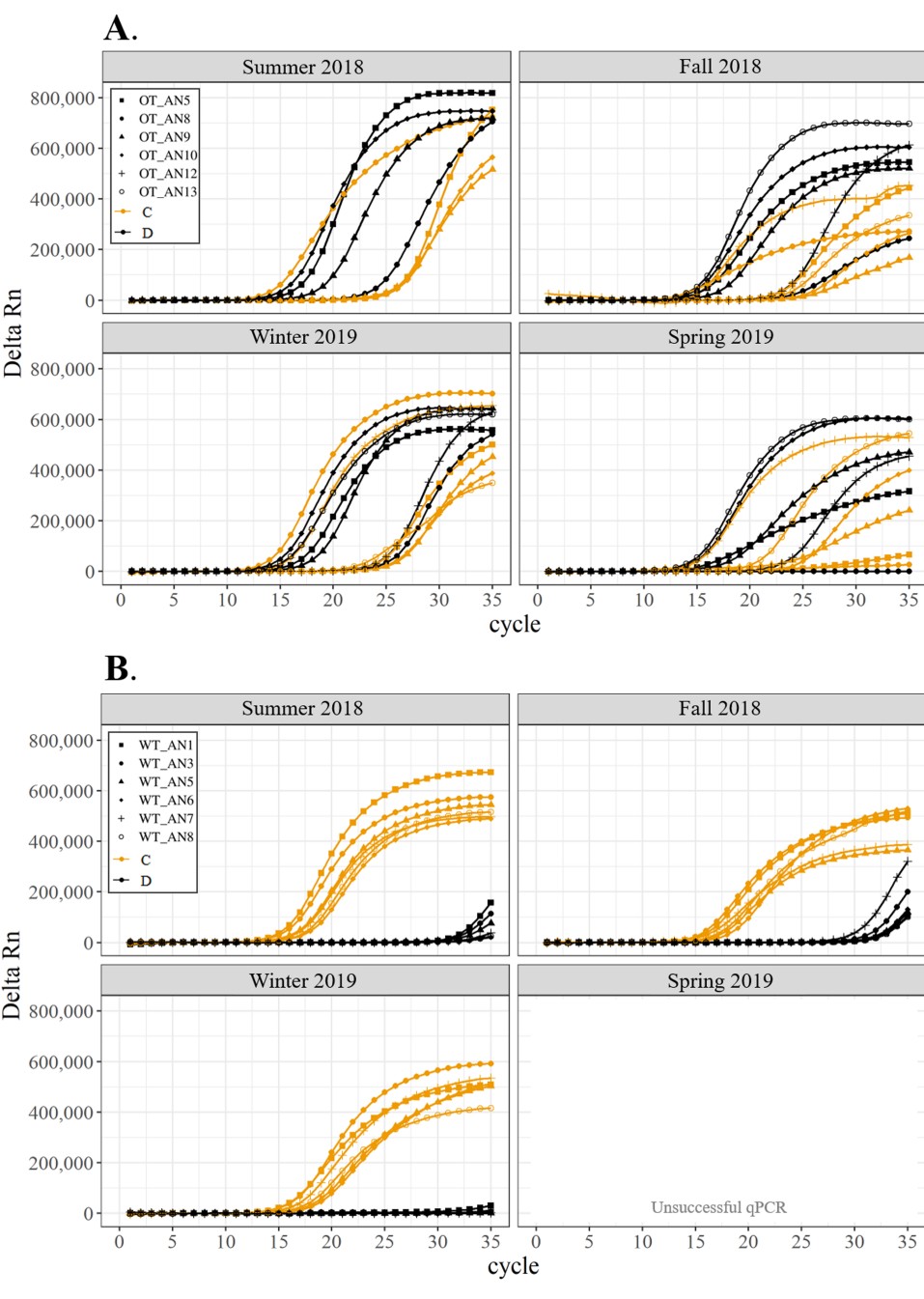

**Figure 2** Detection of the presence or absence of two Symbiodiniaceae genera, *Cladocopium* (orange; C) and *Durusdinium* (black; D), in colonies of *Acropora nana*. Colonies were monitored seasonally at a warmed and thermally variable reef (Outlet reef; panel A) and a thermally stable reef (Wanlitung reef; panel B) in southern Taiwan, from summer 2018 to spring 2019. Summer 2019 samples were not analyzed due to COVID-19 laboratory access limitations. Symbols indicate individual colony data (*e.g.*, in panel A squares show the data for colony #OT_AN5 whereby *Cladocopium* detection is shown in orange and *Durusdinium* detection is shown in black).

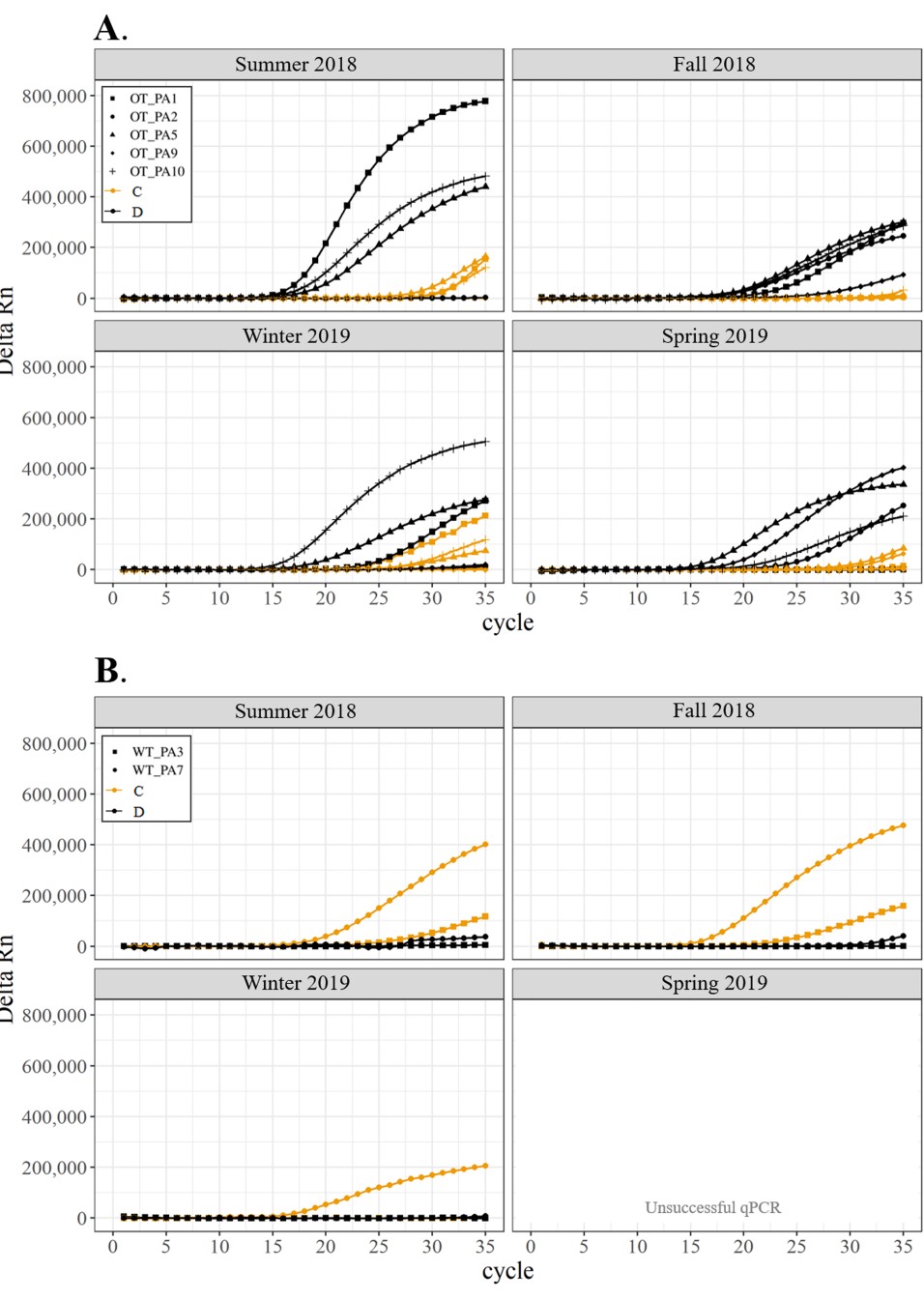

**Figure 3 Detection of the presence or absence of two Symbiodiniaceae genera, *Cladocopium* (orange; C) and *Durusdinium* (black; D), in colonies of *Pocillopora acuta*.** Detection of the presence or absence of two Symbiodiniaceae genera,*Cladocopium* (orange; C) and *Durusdinium* (black; D), in colonies of *Pocillopora acuta*. Colonies were monitored seasonally at a warmed and thermally variable reef (Outlet reef; panel A) and a thermally stable reef (Wanlitung reef; panel B) in southern Taiwan, from summer 2018 to spring 2019. Summer 2019 samples were not analyzed due to COVID-19 laboratory access limitations. Symbols indicate individual colony data (*e.g.*, in panel A squares show the data for colony #OT_PA1 whereby *Cladocopium* detection is shown in orange and *Durusdinium* detection is shown in black).

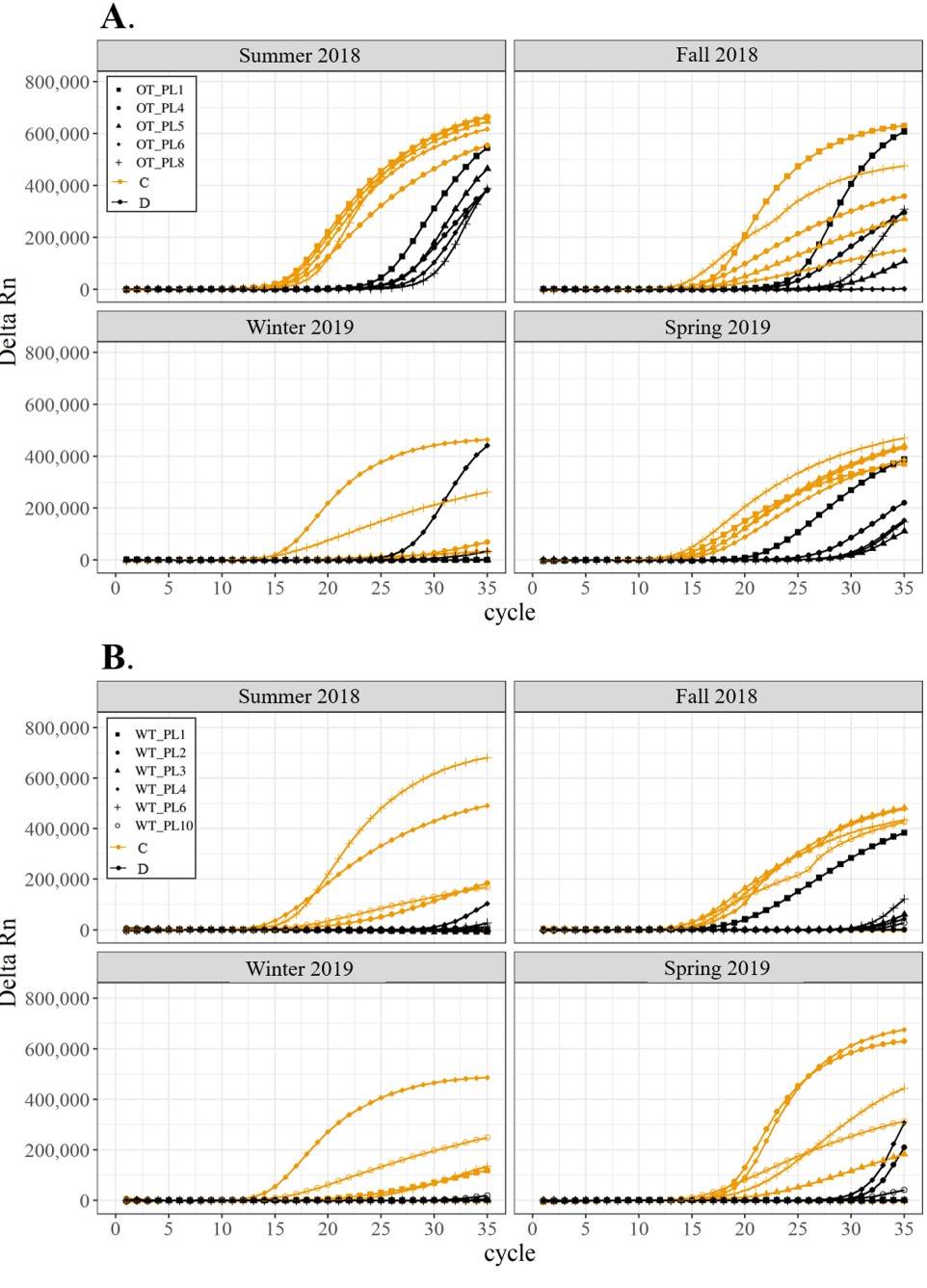

**Figure 4** Detection of the presence or absence of two Symbiodiniaceae genera, *Cladocopium* (orange; C) and *Durusdinium* (black; D), in colonies of *Porites lutea*. Colonies were monitored seasonally at a warmed and thermally variable reef (Outlet reef; panel A) and a thermally stable reef (Wanlitung reef; panel B) in southern Taiwan, from summer 2018 to spring 2019. Summer 2019 samples were not analyzed due to COVID-19 laboratory access limitations. Symbols indicate individual colony data (*e.g.*, in panel A squares show the data for colony #OT_PL1 whereby *Cladocopium* detection is shown in orange and *Durusdinium* detection is shown in black).

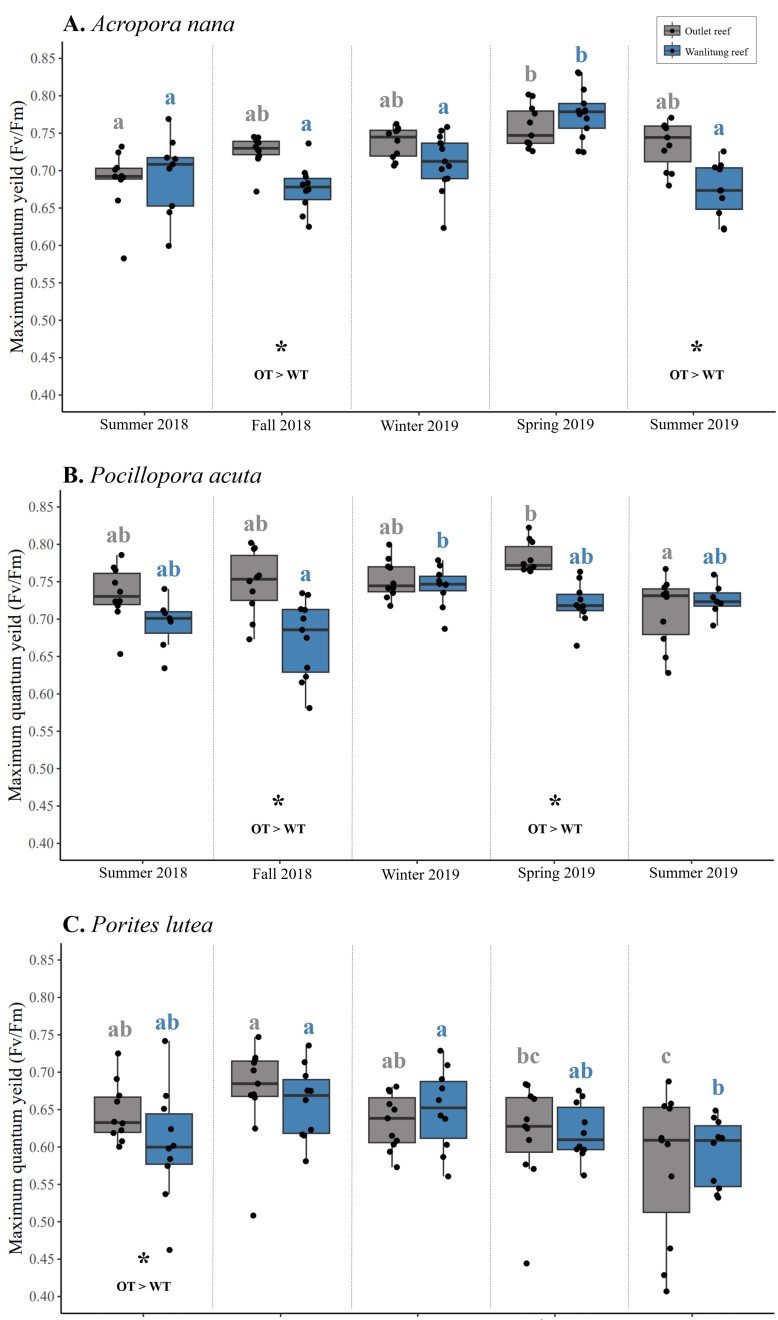

**Figure 5** Maximum quantum yield of photosystem II ($F_v/F_m$) of algal symbionts hosted by three coral species. Colonies of *Acropora nana* (A), *Pocillopora acuta* (B), and *Porites lutea* (C) were monitored at a warmed and thermally variable site, Outlet reef (OT; grey), and a thermally stable site, Wanlitung reef (WT; blue) in southern Taiwan, from summer 2018 to summer 2019. Asterisks indicate significant reef site differences, which was assessed seasonally and independently for each species. Coloured letters indicate significant differences across seasons independently for colonies at Outlet reef (grey letters) and Wanlitung reef (blue letters) separately assessed for each coral species.

relatively consistent across seasons. Reef site patterns in Symbiodiniaceae associations were not clearly mirrored by similar site patterns in photochemical efficiency. Intraspecific $F_v/F_m$ differed between sites in only one or two seasons for each coral species, with corals from the warmed and thermally variable reef periodically having higher $F_v/F_m$. Interspecific differences showed similar patterns at both sites, with *A. nana* and *P. acuta* typically having higher $F_v/F_m$ than *P. lutea*. Although there was some subtle variation in $F_v/F_m$ over time for each species at each site, no consistent seasonal trends in $F_v/F_m$ were observed.

## Temperature and nutrients: reef site comparison

The main physical difference between the two reef sites, based on the parameters assessed in this study, was the thermal regime (Fig. 1). Outlet reef had higher maximum temperatures, particularly in summer months, than Wanlitung reef as the former is chronically influenced by warm-water effluent from an adjacent nuclear power plant. However, Outlet reef also had lower daily temperature minima due to cold-water upwelling in Nanwan Bay (*Hsu et al., 2020*). As upwelling somewhat mitigated the power plant warming, mean daily temperature at Outlet reef was higher than at Wanlitung reef only in winter and spring. Corals that experience high variability or extremes in temperature have been shown to have increased thermal tolerance owing to acclimatization and/or adaptation (*e.g.*, genetic adaptations (*Barshis et al., 2013*), morphological adaptations (*Enríquez et al., 2017*), hosting thermally tolerant algal symbionts (*Oliver & Palumbi, 2011*); but also see *Le Nohaïc et al., 2017*; *Smith et al., 2017*; *Klepac & Barshis, 2020* for limits on adaptation in corals from variable/extreme reefs). In contrast, reefs with relatively stable thermal regimes tend to have corals that are more susceptible to bleaching under elevated temperatures (*Thomas et al., 2018*; *Safaie et al., 2018*).

In contrast to temperature regime patterns, nutrient concentrations at both sites were similar across the study period (Table S5). Levels of $BOD_5$, $NO_3^-$, $NO_2^-$, $NO_3^-$, $NH_3$, and $PO_4^{3-}$ measured across seasons did not differ between Outlet and Wanlitung reefs and were relatively low in comparison to other reefs impacted by anthropogenic stressors in southern Taiwan (*Meng et al., 2008*; *Liu et al., 2012*). Performance patterns of corals between sites are therefore more likely to be attributable to differences in thermal regime than to nutrient levels.

## Symbiodiniaceae dynamics

We found differing patterns in Symbiodiniaceae genera detection between our thermally distinct reef sites (Figs. 2–4). Each of the three coral species showed earlier detection (based on qPCR cycle) of the more thermally tolerant *Durusdinium* algal symbiont at the warmed and thermally variable reef than at the thermally stable reef. The presence/absence (and also the proportional composition) of Symbiodiniaceae genera in coral tissues can change after stress events (*e.g.*, bleaching) with more thermally tolerant genera typically replacing thermally sensitive ones (*Jones et al., 2008*; *Cunning, Silverstein & Baker, 2015*, but see *Kao et al., 2018*; *Rouzé et al., 2019*, for examples of limited shuffling). It is probable that the chronic warming influence, and associated high summer maximum temperatures, at Outlet reef have resulted in a shift to corals potentially hosting more thermally tolerant

Symbiodiniaceae. Indeed, 16 coral genera (including the three genera considered in our study) have been found to predominantly associate with either *Durusdinium* or a combination of *Cladocopium* and *Durusdinium* at Outlet reef (*Keshavmurthy et al., 2014*). In contrast, the same genera at nearby reefs not influenced by the nuclear power plant, and corals deeper than 7 m at Outlet reef and hence out of range of the warm effluent, predominantly host *Cladocopium* (*Keshavmurthy et al., 2014*). Potentially associating with more *Durusdinium* may allow corals at Outlet reef to resist bleaching under high summer temperature maxima and/or facilitate their capacity to recover from bleaching (*Silverstein, Cunning & Baker, 2015*). Corals at Wanlitung reef, like many Indo-Pacific coral species that live under more stable conditions, may tend to preferentially host *Cladocopium* because it is an abundant and species-rich Symbiodiniaceae (*LaJeunesse et al., 2018*) that is not associated with the energetic trade-offs of hosting the more thermally tolerant *Durusdinium* (*e.g.*, *Jones & Berkelmans, 2011*). We did not explicitly quantify dominant or background levels of Symbiodiniaceae genera in this study, but these observed patterns in qPCR detection would likely benefit from a deeper examination (*e.g.*, moving beyond genus-level to species-level assessment is increasingly viewed as important for improving our understanding of coral-algal symbiont dynamics; see *Davies et al., 2023*).

In general, Symbiodiniaceae genera associations remained relatively stable over time for each species and reef site (see also *Epstein et al., 2019*). This seasonal stability reflects the fact that temperatures remained relatively moderate (*i.e.*, within typical site-specific seasonal ranges) throughout our study, with no heatwaves or summer mass bleaching observed. Symbiodiniaceae genus fidelity may also be attributed to acclimatization/adaptation to reef site characteristics (*e.g.*, *Iglesias-Prieto et al., 2004*; *Howells et al., 2020*), long-standing co-evolution of host and symbionts (*Thornhill et al., 2014*; *Turnham et al., 2021*), and/or to a species-specific 'Symbiodiniaceae signature' to the host colony (*Rouzé et al., 2019*).

## Photochemical efficiency dynamics

We examined algal symbiont photochemical efficiency dynamics by tracking $F_v/F_m$ across coral species and seasons at both reef sites (Fig. 5). Values of $F_v/F_m$ usually decrease in response to elevated temperature (*Jones et al., 2000*; *Okamoto et al., 2005*; *Silverstein, Cunning & Baker, 2015*) and can vary seasonally under non-stressed conditions (*Warner et al., 2002*), often as a response to annual fluctuation in solar irradiance (*Winters, Loya & Beer, 2006*). However, we found few intraspecific differences in $F_v/F_m$ between sites, and inconsistent seasonal trends in $F_v/F_m$ among species. Occasional differences between reef sites and seasons were also found by *Carballo-Bolaños et al. (2019)* who showed that $F_v/F_m$ in the brain coral *Leptoria phrygia* was lower at Wanlitung than Outlet reef (in summer and winter, but not spring), and seasonally variable at Wanlitung reef but not at Outlet reef. In our study, typical seasonal differences in temperature and light may not have been enough to elicit significant changes in $F_v/F_m$ for our coral species. Future investigation into other photochemical metrics may be useful to better assess photosystem dynamics (*e.g.*, *Ragni et al., 2010*). Alternatively, clear patterns in $F_v/F_m$ seasonality may have been masked by subtle seasonal fluctuations in Symbiodiniaceae composition, *i.e.*, due to differential photochemical performance across genera (*Kemp et al., 2014*).

We did nevertheless observe relatively consistent interspecific differences in $F_v/F_m$. At our sites, during each season, the $F_v/F_m$ for all coral species was within a typical healthy range for coral species in southern Taiwan (*Putnam, Edmunds & Fan, 2010*; *Mayfield, Fan & Chen, 2013*; *Carballo-Bolaños et al., 2019*), but *A. nana* and *P. acuta* had higher $F_v/F_m$ than *P. lutea* in most seasons. Differences in $F_v/F_m$ among species are not uncommon (*e.g.*, higher $F_v/F_m$ in unstressed *Pocillopora meandrina* compared to *Porites rus* (*Putnam & Edmunds, 2008*); wider thermal breadth in *Porites cylindrica* than *Acropora valenciennesi* (*Jurriaans & Hoogenboom, 2020*)). This is likely a result of taxon-specific traits, such as coral tissue thickness (*Anthony & Hoegh-Guldberg, 2003*), algal symbiont position within the coral tissue (*Edmunds, Putnam & Gates, 2012*), and/or genus-specific symbiont associations (*Wang et al., 2012*; *Yuyama et al., 2016*).

## CONCLUSIONS

We did not detect clear evidence of seasonal trends in dominant Symbiodiniaceae genera or photochemical efficiency in our study species, rather differences were more apparent among sites with contrasting thermal regimes and among coral species—highlighting the importance of species-specific studies. Reef site patterns that we observed in Symbiodiniaceae genera detection, using a coarse presence/absence qPCR approach, merit more comprehensive investigation (*i.e.,* genera or species-level quantification) to better assess the influence of thermal regime on Symbiodiniaceae associations among coral hosts. Baseline seasonal data under non-stressed conditions are pertinent to improve our understanding of energy provision sources relevant to coral thermal tolerance. Identifying typical ranges of normal variability in coral and Symbiodiniaceae physiology, coupled with an appreciation of the role that species traits and reef characteristics play, will allow for a better understanding of coral holobiont resistance and resilience in a warming ocean.

## ACKNOWLEDGEMENTS

Thank you to Jing-Ya Yan for undertaking the water chemistry analysis and Tai-Chi Chang for dive assistance. The manuscript was improved by feedback from Manon Picard, Jillian Dunic, Hannah Watkins, and Helen Yan.

### Funding

Funding was provided by the Taiwan Ministry of Science and Technology and was awarded to Tung-Yung Fan (MOST 107-2611-M-291-004). The funders had no role in study design, data collection and analysis, decision to publish, or preparation of the manuscript.

### Grant Disclosures

The following grant information was disclosed by the authors:
The Taiwan Ministry of Science and Technology and was awarded to Tung-Yung Fan: MOST 107-2611-M-291-004.

## Competing Interests

The authors declare there are no competing interests.

## Author Contributions

- Crystal J. McRae conceived and designed the experiments, performed the experiments, analyzed the data, prepared figures and/or tables, authored or reviewed drafts of the article, and approved the final draft.
- Shashank Keshavmurthy conceived and designed the experiments, performed the experiments, analyzed the data, prepared figures and/or tables, authored or reviewed drafts of the article, and approved the final draft.
- Hung-Kai Chen performed the experiments, authored or reviewed drafts of the article, and approved the final draft.
- Zong-Min Ye performed the experiments, authored or reviewed drafts of the article, and approved the final draft.
- Pei-Jie Meng performed the experiments, authored or reviewed drafts of the article, and approved the final draft.
- Sabrina L. Rosset conceived and designed the experiments, performed the experiments, authored or reviewed drafts of the article, and approved the final draft.
- Wen-Bin Huang conceived and designed the experiments, authored or reviewed drafts of the article, and approved the final draft.
- Chaolun Allen Chen conceived and designed the experiments, authored or reviewed drafts of the article, and approved the final draft.
- Tung-Yung Fan conceived and designed the experiments, authored or reviewed drafts of the article, and approved the final draft.
- Isabelle M. Côté conceived and designed the experiments, authored or reviewed drafts of the article, and approved the final draft.

## Field Study Permissions

The following information was supplied relating to field study approvals (i.e., approving body and any reference numbers):

Coral collection permits for this study were provided by Kenting National Park (Permit numbers: 1080000091, 1090006457).

## Data Availability

All data and R scripts used in our analyses are available at GitHub and Zenodo:
https://github.com/CJ-McRae/McRae-et-al_Peer-J-submission

Crystal McRae. (2023). CJ-McRae/McRae-et-al_Peer-J-submission: McRae et al. PeerJ submission (Version v1). Zenodo. https://doi.org/10.5281/zenodo.7762107.

## Supplemental Information

Supplemental information for this article can be found online at http://dx.doi.org/10.7717/peerj.15421#supplemental-information.

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
