# Peer review of "Baseline dynamics of Symbiodiniaceae genera and photochemical efficiency in corals from reefs with different thermal histories"

_PeerJ, doi:10.7717/peerj.15421_

## Round 0.1 · original submission · Major Revisions

First, let me apologize for the time it has taken to make a decision on this manuscript. I have now received comments from three expert reviewers who have all provided a thorough list of points to consider.

Overall, this manuscript is well-written and clear and contains a very useful data set. However, there are a number of issues that must be attended to if you decide to resubmit this manuscript. Some of the concerns are with the methods where clarification is needed on the qPCR methods and ensure that you include the raw qPCR data when uploading your resubmission. Another important issue that was raised was to clarify the statistical analysis and data interpretation with respect to the dominant symbiont genera. Other important points have also been brought up and all should be considered in a revision of the manuscript. Please ensure you follow the journal guidelines when preparing your rebuttal and revised version of the manuscript.

Reviewer 1 ·

Basic reporting

The manuscript is well-written. Very clear and concise. The work is described in sufficient detail. And the relevance and outcomes are put into a larger context. In general, the methods and analysis seem adequate and the results support the general inferences. I was in favor of publication after minor revisions, but one section of the analysis (related to figure 4) requires a more in-depth review of the statistical analysis and interpretation of the data.
Data: Data and code are provided in a repository. Thank you.
Figures: Fig 3. I suggest adding a color legend to the figure itself. I know the colors are explained in the figure legend, but readers like me might want to be able to interpret the figure at glance without needing the read the whole legend text.
Fig 4. I think this figure and analysis are misleading. See below.

Experimental design

The research problem is defined and the study helps to understand better the seasonal dynamics of Symbiodiniaceae communities. The methods are described in detail. Please see the comments on the validity of the findings.

Validity of the findings

I am happy with the analyses used in the study, with the exception of the one used to test the effect of the dominant symbiont genus on Fv/Fm. In this analysis, all data by dominant symbiont is pooled, which I consider ok in the case of the colony (inside an spp) and season. However, pooling the data of all species to assess the effects of the dominant symbiont genus is not adequate since the types of symbiont communities are strongly dominated by one coral species (i.e., all Durusdinium samples correspond to P. acuta, most D>>C correspond to A. nana, and most C>>D correspond to P. lutea). As a consequence of these unbalanced groups, the coral species are a huge confounding factor (i.e., you can't tell if a higher Fv/Fm is the result of the Symbiodiniaceae genus or the coral species itself). I suggest either running this analysis by coral species (if there are enough samples to divide them into multiple symbiont categories) or removing it since it can be misleading.
Running a more robust analysis for this question is necessary since previous studies have found the opposite results with respect to the effects of Symbiodiniaceae genera on Fv/Fm values under non-stressful temperatures (i.e., colonies of the same species that host Durusdinium often have lower Fv/Fm than colonies hosting Brevioulm or Cladocopium) and have also highlighted that different coral species often have different Fv/Fm (see 10.1007/s00338-017-1640-3, 10.1007/s00338-021-02159-x for example).

Additional comments

Results: I suggest o include a brief description of the average nutrient values in the results section. In the discussion, there is mention of relatively low nutrients in the study sites compared to other impacted reefs, but the reader cannot have an idea of what these values are without looking at the supplementary information.

Line 640: Fix the typo

Supplementary info: Please add legends to the supplementary figures and tables.

Reviewer 2 ·

Basic reporting

Overall, the ms by McCrae and colleagues provides an interesting temporal dataset of environmental conditions and symbiosis associations of reef-building corals from reef sites with contrasting thermal histories in Taiwan. Though I address concerns in my review, I think this will be a suitable contribution for PeerJ following further clarification on the qPCR methods and revising language surrounding Symbiodiniaceae nomenclature/ecology.

Major concern surrounding data availability: Raw data for qPCR (Delta RN) and environmental parameters through time is missing (update- environmental data is provided on the associated GitHub but raw qPCR data was not).

Figures: images in the main text, uploads, and supplement appear granulated. Perhaps these can be saved at a higher level of resolution and re-uploaded.

Supplemental tables: I was unable to locate captions specifying acronym meanings

Major concerns surrounding language used for Symbiodiniaceae nomenclature and ecology.
1. Several of the Symbiodiniaceae binomial species designations can be inferred from the literature. For example, corals from the general Pocillopora and Montipora associate with Durusdinium glynii.
a. Refs (D. glynnii): Wham et al 2017 Phycologia https://www.tandfonline.com/doi/full/10.2216/16-86.1, Johnston et al 2022 Mol Ecol https://onlinelibrary.wiley.com/doi/epdf/10.1111/mec.16654
b. Refs (Poc. acuta associations with Cladocopium pacificum but sometimes C. latusorum): Johnston et al 2022 Mol Ecol; Turnham et al 2021 ISMEJ https://www.nature.com/articles/s41396-021-01007-8
c. Ref (Porites associations with Cladocopium C15): Hoadley et al 2021 Global Change Bio https://onlinelibrary.wiley.com/doi/full/10.1111/gcb.15799; Forsman et al 2020 Sci Rep https://www.nature.com/articles/s41598-020-73501-6
2. Incorporating species names of both host coral and algal symbiont entities is vital to ensuring proper differentiation of the physiological ecology of holobiont pairings in the literature. In other words, broad-sweeping conclusions assigned at the symbiont (or host) genus level overlooks complexities in holobiont resilience that are explained by host-symbiont pairings (ex, Hoadley et al 2019 Scientific Reports). Please see section 1.1 (page 7) of Davies et al 2022 “Building Consensus around the Assessment and Interpretation of Symbiodiniaceae Diversity” (https://www.preprints.org/manuscript/202206.0284/v1) for a more detailed description of why species-level nomenclature is key to pushing the field forward and how genus-level generalizations overlook exciting complexities observed in coral symbiosis. I recommend revising the intro paragraphs starting at lines 78 and 93.

Experimental design

Major concerns regarding qPCR methods.
1. Were cell culture lines or a similar positive control used for verifying the presence of Cladocopium versus Durusdinium? See Correa et al 2009 Marine Biology https://link.springer.com/article/10.1007/s00227-009-1263-5
2. Along these lines, was the extensive difference in Cladocopium vs Durusdinium copy number variation accounted for when ascertaining the presence/absence of each genus? More information is needed about how dominant genus was assigned. See Saad et al 2020 Frontiers (https://www.frontiersin.org/articles/10.3389/fmicb.2020.00847/full) and LaJeunesse et al 2008 L&O (https://aslopubs.onlinelibrary.wiley.com/doi/abs/10.4319/Lo.2008.53.2.0719)
3. The Abstract (lines 54-55) and results (subsection at line 271) insinuates that relative abundance (rather than presence/absence) was ascertained for the data which conflicts with the presence/absence assay described in the methods (line 199). The application of 30 pcr cycles as a threshold for detecting a ‘background symbiont’ does not seem justified given the lack of positive control and widespread copy number variation among Cladocopium/Durusdinium

Lines 227-231 and 287: Thank you for your transparency!

Validity of the findings

Overall, I think the ms provides an interesting temporal dataset with sufficient replication.

Specific comments on conclusions from the discussion section-
Line 372: Is there any information on light regimes and carbon chemistry across these sites? I appreciate the synthesis with biogeochemistry literature but light and carbon chemistry are also critical drivers of symbiont performance. See Lopez et al 2022 Scientific reports for more on light as a driver of biodiversity https://www.nature.com/articles/s41598-022-25094-5

Paragraph at line 398: long-standing co-evolution ought to be mentioned here (ex, Turnham et al 2021 ISMEJ https://www.nature.com/articles/s41396-021-01007-8 and Thornhill et al 2013 Evolution https://onlinelibrary.wiley.com/doi/full/10.1111/evo.12270)

Photochemical efficiency dynamics paragraph (line 406): I appreciate how the ms’ discussion does not overinterpret Fv/Fm data as a sole metric for “photosynthetic health”. Here I think it could be mentioned that future evaluation of other aspects of photophysiology more ‘downstream’ of PSII photochemical efficiency may provide more insight into photosystem dynamics and carbon fixation as a whole (ex: Ragni et al 2010 MEPS https://www.int-res.com/articles/meps2010/406/m406p057.pdf)

·

Basic reporting

Extremely well done

Experimental design

Great

Validity of the findings

Great

Additional comments

One of the easiest reviews I've ever done as the paper was so well written. A few minor comments where I was confused below:

Methods

L166: I didn't have access to the supplementary information (not sure if that's my fault or not), but I think it could be relatively easy here to mention in the main text how many new colonies were tagged and sampled since you said it was a rare occurrence.

L183-5: I don't know what you mean here about the subsets for each season and their exceptions. For instance, you said "only n=2 colonies/season" for P. acuta from Wanlitung reef, but looking at figure 2 there's also an n of 1 in the last time point? & I think what you were saying for exceptions is just the ones that didn't have 6 colonies, but can you please make that more clear?

L186: Parentheses should go "Ferrara et al. (2006)" I think

L208-10: The font changes on some of the Celsius 'C's for some reason

L216: Can you mention more geographical location for the aquarium (city, country, etc.) as I'm not familiar with this area

L224: italicize 'in situ' & other Latin derived phrases

Results & throughout the manuscript: italicize 't' & 'p' values for stats (that's the standard in my area, not sure if it differs elsewhere)

Discussion: throughout, please mention the specific figures you're referring to for your findings just so it's crystal clear. For example (Figure 2) on line 342 or whichever figure(s) it was you're discussing, including supp. figures where appropriate

Figure 1 caption: I don't know if I missed the explanation somewhere, but why is 30˚C the reference & why does it have a dashed horizontal line?

Figure 2 caption: when you say "more abundant genus" how did you define that when you were previously discussing presence/absence & both are present - I just need a little more clarity about what that means in the methods as I'm not familiar with qPCR [e.g. higher concentration? or a brighter/bigger band on the gel?]

Also figure 2 caption: I would mention here that summer 2019 samples were not able to be processed, in case someone skimming through figures is confused about why the NAs are there

---

## Round 0.2 · Minor Revisions

Thank you for resubmitting your manuscript. I have done a quick review and have noticed that your marked up version and clean version do not match. Line 141 (This study was initially reported ...) is not present in the clean version (it would be between lines 128 and 130. Please upload the correct clean version. Also, please check the new text as I found the following errors in the marked up version:

line 53 there is an unnecessary "and"

line 223 insert "the" between on and protocol

line 227 there should be a space between Cto (to read C to) and Cin (to read C in)

I can also provide you with an update to the Davies et al. 2022 reference. It is now Davies et al. 2023 (no change in title) PeerJ DOI: 10.7717/peerj.15023

---

## Round 0.3 · accepted · Accept

I am satisfied with the changes made to the manuscript and am happy to recommend your manuscript be published in PeerJ.